

# Adoption of CRISPR-Cas for crop production: present status and future prospects

Akinlolu Olalekan Akanmu[1], Michael Dare Asemoloye[1], Mario Andrea Marchisio[2] and Olubukola Oluranti Babalola[1]

[1] Food Security and Safety Focus Area, Faculty of Natural and Agricultural Sciences, North-West University, Mmabatho, South Africa
[2] School of Pharmaceutical Science and Technology, Tianjin University, Tianjin, China

## ABSTRACT

**Background**. Global food systems in recent years have been impacted by some harsh environmental challenges and excessive anthropogenic activities. The increasing levels of both biotic and abiotic stressors have led to a decline in food production, safety, and quality. This has also contributed to a low crop production rate and difficulty in meeting the requirements of the ever-growing population. Several biotic stresses have developed above natural resistance in crops coupled with alarming contamination rates. In particular, the multiple antibiotic resistance in bacteria and some other plant pathogens has been a hot topic over recent years since the food system is often exposed to contamination at each of the farm-to-fork stages. Therefore, a system that prioritizes the safety, quality, and availability of foods is needed to meet the health and dietary preferences of everyone at every time.

**Methods**. This review collected scattered information on food systems and proposes methods for plant disease management. Multiple databases were searched for relevant specialized literature in the field. Particular attention was placed on the genetic methods with special interest in the potentials of the Clustered Regularly Interspaced Short Palindromic Repeats (CRISPR) and Cas (CRISPR associated) proteins technology in food systems and security.

**Results**. The review reveals the approaches that have been developed to salvage the problem of food insecurity in an attempt to achieve sustainable agriculture. On crop plants, some systems tend towards either enhancing the systemic resistance or engineering resistant varieties against known pathogens. The CRISPR-Cas technology has become a popular tool for engineering desired genes in living organisms. This review discusses its impact and why it should be considered in the sustainable management, availability, and quality of food systems. Some important roles of CRISPR-Cas have been established concerning conventional and earlier genome editing methods for simultaneous modification of different agronomic traits in crops.

**Conclusion**. Despite the controversies over the safety of the CRISPR-Cas system, its importance has been evident in the engineering of disease- and drought-resistant crop varieties, the improvement of crop yield, and enhancement of food quality.

Corresponding author
Olubukola Oluranti Babalola, olubukola.babalola@nwu.ac.za

## INTRODUCTION

The world's population is estimated to reach ten billion by 2050; hence, a rise in food production by 60 to 100% would be required to support the demand of the growing global population in the near future (*Chen et al., 2019*; *Francisco-Ribeiro & Camargo-Rodriguez, 2020*). The second sustainable development goal (SDG) of the United Nations aims at ending malnutrition and hunger by 2030 as well as ensuring access for everyone to sufficient and nutritious food all year round (*Fanzo, 2019*). Despite concerted efforts to improve the global food system, current agricultural production is still struggling with the challenge of meeting the required level of productivity needed to feed 10 billion people by 2050 (*Hickey et al., 2019*). Further, the effect of biotic (*e.g.*, insect pests, fungi, bacteria, and viruses) and abiotic (drought, heat, salinity, and frost) stressors is heightened with excessive anthropogenic activities. These are causing a decrease in agricultural lands, adequate water resources, and increased competition for the depleting resources, which are recognized as the foremost challenges affecting the productivity of plant products in this age (*Akanmu et al., 2021*; *Babalola, Berner & Amusa, 2007*; *Razzaq et al., 2019*; *Vermeulen, Campbell & Ingram, 2012*). Therefore, achieving a secure and safe food system in the face of an accelerating food demand will be imperative (*Chukwuka et al., 2020*; *Mamphogoro, Babalola & Aiyegoro, 2020*).

Crop improvement aims at increasing the yield, quality, and nutrient values as well as improving resistance against different biotic and abiotic stressors. Today, genetic improvements in food crops have been shown as promising measures for attaining the dietary needs of the increasing population, while safeguarding the preferences and health of individuals who are the end-users of such food. Genetic engineering systems include mega-nucleases, transcription activator-like effector nucleases (TALENs), zinc finger nucleases (ZFNs), and CRISPR-Cas9 with its orthologs (*Jinek et al., 2012*; *Zetsche et al., 2015*; *Pant et al., 2022*). The CRISPR-associated proteins (Cas9 and Cas12a), which are the focus of this review, have been reported as accurate, convenient, and efficient genome editing tools that have opened up opportunities for applications in various fields (*Macovei et al., 2018*; *Mohanraju et al., 2016*; *Swartjes, Staals & van der Oost, 2020*; *Zhang et al., 2018a*; *Zhang et al., 2017*). The CRISPR-Cas9 system has received massive attention as a result of its wide range of usage in plant breeding for the development of agricultural crops and biological research. In particular, genome editing for traits of interest such as disease resistance (*Katoch et al., 2020*), drought tolerance (*Shi et al., 2017*), and salt tolerance (*Zhang et al., 2019*), in major crops such as wheat (*Liang, Chen & Gao, 2019*), maize (*Barman et al., 2019*), and soybean (*Chilcoat, Liu & Sander, 2017*), have been investigated. Thus, this review discusses the contribution and efficiency of CRISPR-Cas9 in improving the security, quality, and safety of the food system.

### Survey methodology

A comprehensive investigation of published articles on CRISPR-Cas applications in plant improvements toward sustainable crop production was conducted. This was carried out through inclusiveness and impartial investigations of literature in line with the method of *De Souza & Bonciu (2022)*, multiple databases including Web of

Science (http://apps.isiknowledge.com), Scopus (http://www.scopus.com/), ScienceDirect (http://www.sciencedirect.com/), and PubMed (http://www.ncbi.nlm.nih.gov/pubmed) and Google Scholar (https://scholar.google.com/) were searched while relevant and specialized literature with the latest publication in the field was further hand searched using the following keywords; Crispr-Cas, genome editing, food security, food safety, food system, sustainable agriculture, and crop production. Without bias, the search results gathered were compiled by employing the online endnote library system to arrange the useful articles embedded in the context. Furthermore, we appraised the titles, abstracts, and the conclusion of the literature to determine the useful ones.

## CRISPR-CAS: THE FUTURE OF THE FOOD SYSTEM

CRISPR-Cas is a cutting-edge technology that can be adopted for plant breeding techniques, it allows making precise cuts in a plant's genome to insert/delete genes for promoting crop development. The subsequent repair of the cut by the cell's endogenous repair mechanisms can introduce precise changes in the targeted chromosomes (*Wu et al., 2018*). The CRISPR-Cas technology may utilize crops' natural traits and does not introduce new genes in the event of gene disruption (*Soda, Verma & Giri, 2018*). Current scientific advancements have shown CRISPR-Cas systems especially type II CRISPR-SpCas9 from *Streptococcus pyogenes* and type V CRISPR-LbCas12a from *Lachnospiraceae bacterium* as a versatile technology whose prospects are yet to be thoroughly mined in human biology (*Li et al., 2023a*; *Li et al., 2023b*), agriculture, and microbiology (*Brandt & Barrangou, 2019*; *Tomlinson, 2018*; *He et al., 2023*). With a focus on food security and safety, the effectiveness of CRISPR-Cas systems and other gene editing techniques lies in the ability to keep established croplands productive in the face of a changing climate and conserve the remaining wild areas of the planet. Gene-editing technologies can reduce farmers' reliance on fertilizers through the development of designer microbes that produce nitrogen for crop use. In addition, CRISPR-Cas systems would help create crops with novel traits that offer more incremental advances in efficiency, sequester more carbon, pack in more nutrients, and produce more food per acre with fewer inputs (*Brandt & Barrangou, 2019*). To adopt this technology, a single-guide RNA (sgRNA) is designed to target a gene of interest, with the aid of CRISPR/Cas9 used to deliver the genetic information into the plant cell, often done *via* suitable bacteria delivery system like the *Agrobacterium*-mediated transformation method/ribonucleoprotein (RNP). These commonly involve (i) tissue culture for callus induction, (ii) plant regeneration from CRISPR/Cas9-mutated tissues, (iii) generation of T0 CRISPR/Cas9-mutated transgenic plants, (iv) screening of transgenic plants to detect on- and off-target efficiency of CRIPR-Cas9, (v) self-pollination of $T_0$ transgenic plants for generation of homozygous $T_1$ plants, till (vi) phenotypic analysis of $T_1$ plants (Fig. 1).

The introduction of small indels or premature stop codons that result in gene knockouts or null alleles causes frame-shift mutations, which has been the most frequent strategy employed in CRISPR-Cas application (*Zhang et al., 2018a*). Thus, the technology over the years has been rapidly exploited in plant improvements and it poses an effective solution to many problems in plant breeding (*Gao, 2018*; *Ricroch, Clairand & Harwood, 2017*;

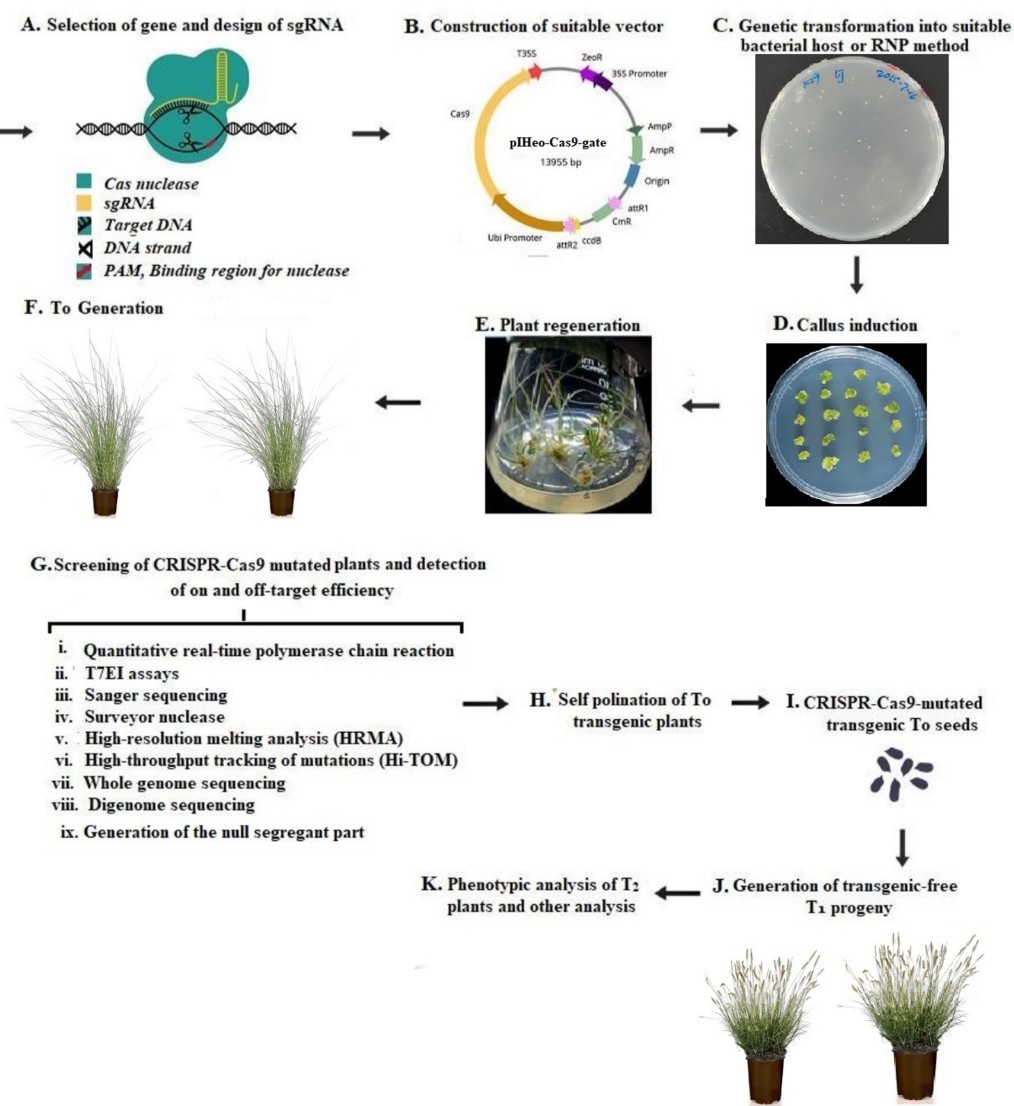

**Figure 1** **Workflow for the use of CRISPR-Cas9-based gene editing in plants.** (A) Selection of the target gene for plant transformation and designing of single-guide RNA (sgRNA) for the target gene. (B) construction of suitable vector, (C) genetic transformation for the delivery of CRISPR/Cas9 *via* suitable bacteria delivery system like the Agrobacterium mediated transformation method/ribonucleoprotein (RNP), (D) tissue culture for callus induction, (E) plant regeneration from CRISPR/Cas9-mutated tissues, (F) generation of $T_0$ CRISPR/Cas9-mutated transgenic plants, (G) screening of transgenic plants to detect on- and off-target efficiency of CRIPR-Cas9,by detection of on- and off-target efficiency by Sanger sequencing, (H) self-pollination of $T_0$ transgenic plants for generation of homozygous $T_1$ plants, (I) generation of CRISPR/Cas9-mutated $T_0$ Seeds, (J) generation of transgene-free $T_1$ progeny, (K) phenotypic analysis of T1 plants and other analysis.

*Zhang, Li & Li, 2016*). It has also been used to create genetically engineered plants, which would significantly boost agricultural yields and other important features in commercially genetically modified crops (*Stout, Klaenhammer & Barrangou, 2017*; *Shi et al., 2017*; *Zhang et al., 2018a*; *Verma et al., 2023*). It is perceived to have an advantage over traditional

breeding methods, in that it enables the researchers to generate suitable germplasms such as disease resistance, herbicide resistance, and improved yield and quality, which occur by the removal of undesirable genetic elements or inserting gain-of-function mutations *via* precise genome editing. This has been established for the improvement of several plants including wheat, cassava, tomatoes, and corn (*Castiglioni et al., 2008*; *Francisco-Ribeiro & Camargo-Rodriguez, 2020*; *Khatodia, Bhatotia & Tuteja, 2017*; *Shi et al., 2017*; *Zaidi et al., 2016*). Some of its applications currently under study show features such as improved biotic and abiotic values, as in the development of resistance to powdery mildew disease in wheat and tomato, resistance to viruses in potato and cucumber, and drought tolerance in soybean among others (Table 1). Applications of CRISPR-Cas have also enabled, in some crops, the improvement of yield traits and nutrient qualities such as the increased protein and amylose contents in wheat, enhancement of lycopene contents, fruit color, shape, and longer shelf-life in tomato, the breeding for lipoxygenase-free soybean and production of cyanide-free cassava (Table 2). Similarly, CRISPR-Cas is instrumental in increasing the fruit size, grain length, width, and weight as observed in wheat, tomato, groundcherry, and rice, while it has also been employed in the spreading of plant tillers in wheat (Table 3). However, most engineered crops are at different stages of research, development, and production, with many of them awaiting regulatory approvals (*Chen et al., 2019*).

## CROP IMPROVEMENT USING CRISPR-CAS AS GENETIC ENGINEERING TOOL

Today, there is a wide availability of genomic information on varying plant species and this development could be linked to the advancement in sequencing technologies that serve as the gateway to precision gene editing (*Ahmar et al., 2020*). Modern breeding strategies for the improvement of crop plants employ the deciphering of numerous biological mechanisms and elucidation of the role of genetic and epigenetic factors (*Gallusci et al., 2017*; *Richardson, Kelsh & Richardson, 2023*). The hallmark of modern genetic engineering rests on the ability to couple specific sequences to induce a break in the DNA. When combined with a desired DNA binding domain (DBD), a nuclease may be directed to a specific target location (*Ranjha, Howard & Cejka, 2018*). DNA cleavage, then, activates the host's DNA repair system, resulting in mutations and the probable loss of function of a specific gene of interest (*Jinek et al., 2012*; *Zetsche et al., 2015*; *Globus & Qimron, 2018*). The recombinant DNA technology, therefore, allows the transfer of the desired genes from any plant or microorganism into crops. This, subsequently, produces new genotypes and phenotypes that offer the possible yield enhancement required for breeding purposes and ultimately improve fruit/crop quality (*Zhang et al., 2019*; *Das et al., 2023*). Homing meganucleases, zinc finger nucleases, transcription activator-like effector nucleases, and CRISPR-Cas9 and -Cas12a orthologs are the four types of designed nucleases created for genome engineering. The first three approaches, however, have a closed architecture that makes manipulating target specificity difficult, time-consuming, and expensive (*Guha & Edgell, 2017*; *Chandran et al., 2023*).

**Table 1** The CRISPR-modified crops with improved biotic and abiotic characteristics.

| Crop species | Target gene | Associated Trait | Reference |
|---|---|---|---|
| Wheat (*Tritocum aestivum*) | EDR1 | Developing resistance to powdery mildew disease caused by *Blumeria graminis* f.sp. (Btg) Tricitici | *Zhang et al. (2017)* |
| | OsERF922 | Resistance to rice blast disease | *Wang et al. (2016)* |
| | OsSWEET13 | Resistance to bacteria blight | *Zhou et al. (2015)* |
| | ALS, EPSPS | Resistance to herbicide | *Butt et al. (2017)*; *Endo, Mikami & Toki (2016)*; *Li et al. (2016)*; *Sun et al. (2016)* |
| Tomato (*Solanum lycopersicum* L.) | SlMLO1 | Resistance to powdery mildew disease | *Nekrasov et al. (2017)* |
| | SlJAZ2 | Resistance to bacteria speck disease | *Ortigosa et al. (2019)* |
| Maize (*Zea mays* L.) | ALS | Herbicide resistance | *Svitashev et al. (2015)* |
| | ARGOS8 | Tolerance to drought | *Shi et al. (2017)* |
| Potato (*Solanum tuberosum* L.) | ALS | Herbicide resistance | *Butler et al. (2016)* |
| | eIF4E and eIF(iso)4E | Resistance against viruses and cold-induced sweetening | *Hameed et al. (2020)* |
| Grapefruit (*Citrus paradisi*) | CsLOB1 promoter | Resistance to citrus canker disease | *Jia et al. (2016)*; *Jia et al. (2017)* |
| Orange (*Citrus sinensis*) | CsLOB1 promoter | Resistance to citrus canker disease | *Peng et al. (2017)* |
| Cucumber (*Cucumis sativus*) | eIF4E | Resistance to viral disease | *Chandrasekaran et al. (2016)* |
| Soybean (*Glycine max* [L.] Merr. | ALS | Herbicide resistance | *Li et al. (2015)* |
| | HaHB4 | Drought tolerance | *Martignago et al. (2020)* |
| Cassava (*Manihot esculenta* Crantz) | EPSPS | Herbicide resistance | *Hummel et al. (2018)* |
| | eIF4E isoforms nCBP-1 and nCBP-2 | Reduction in cassava brown streak disease symptom severity and incidence | *Gomez et al. (2019)* |
| Flax (*Camelina sativa*) | EPSPS | Herbicide resistance | *Sauer et al. (2016)* |
| Kale (*Brassica oleracea*) | CRTISO | Yellow leaves and stems | *Sun et al. (2020)* |

The CRISPR-Cas systems, rather than protein-DNA complexes, use guide-RNA (gRNA) and ribonucleotide-DNA complexes for target identification and subsequent cleavage, modification, deletion, insertion, or replication, which makes them more flexible to use (*Jinek et al., 2012*; *Zetsche et al., 2015*; *Globus & Qimron, 2018*). CRISPR-Cas9 and CRISPR-Cas12a are most widely used in plants. The elucidation of the biochemical mechanism of the CRISPR-Cas9 system, for instance, was first reported in 2012 and since then has revolutionized genetic research in life sciences. They can both target DNA, carry out a double-strand break (DSBS), and have therefore been employed in delivering DNAs, RNAs, or even protein-RNA into active site-directed nucleases (SDN) in plant cells (*Zhang et al., 2018a*; *Zhang et al., 2017*).

CRISPR–Cas has emerged as a valuable tool in agriculture due to its unequaled ability to accurately change plant genomes (*Khan et al., 2022*). It has changed present breeding processes and assisted in the development of novel varieties of crops with desirable traits

**Table 2   The CRISPR-modified crops with improved yield quality.**

| Crop species | Target gene | Associated Trait | Reference |
|---|---|---|---|
| | *OsMATL* | Inducing haploid plants | *Yao et al. (2018)* |
| | *TaGW2, α-gliadin genes* | Low gluten content | *Sánchez-León et al. (2018)*; *Wang et al. (2018)* |
| Wheat (Tritocum aestivum) | *SBEIIb* | Increased amylose content | *Sun et al. (2017)* |
| | *GW2* | Increase in protein content | *Zhang et al. (2018a)* |
| | *SP, SP5G, CLV3, WUS, GGP1* | Domestication of tomato | *Li et al. (2018a)* |
| | *SlAGL6* | Parthenocarpy | *Klap et al. (2017)* |
| | *ANT1* | Fruit color (purple) | *Čermák et al. (2015)*; *Vu et al. (2020)* |
| | *SlMYB12* | Fruit color (pink) | *Deng et al. (2018)* |
| | *CRTISO* | Fruit color (tangerine) | *Ben Shlush et al. (2020)* |
| | *Psy1, CrtR-b2* | Fruit color (yellow) | *D'Ambrosio, Stigliani & Giorio (2018)* |
| Tomato (Solanum lycopersicum L.) | *OVATE, Fas, Fw2.2* | Fruit size, oval fruit shape | *Zsögön et al. (2018)* |
| | *TaGW7* | Grain shape | *Wang et al. (2019)* |
| | *ALC* | Long shelf life | *Yu et al. (2017)* |
| | *PL, PG2a, TBG4* | Long shelf life | *Uluisik et al. (2016)* |
| | *slyPDS* | Increased lycopene content | *Li et al. (2018b)* |
| | *BnFAD2* | High oleic acid proportion | *Okuzaki et al. (2018)* |
| | *TMS5* | Thermosensitive male-sterile | *Liu et al. (2017)* |
| Maize (Zea mays L.) | *Psy1* | Seed color | *Zhu et al. (2016)* |
| | *Wx1* | Waxy corn | *Gao et al. (2020)* |
| | *StGBSS* | Low amylose content | *Veillet et al. (2019)* |
| | *StSBE1, StSBE2* | High amylose content | *Tuncel et al. (2019)* |
| Potato (*Solanum tuberosum* L.) | *OsBEI and OsBEIIb* | High amylose content | *Sun et al. (2017)* |
| | *CrtI, PSY* | High β-carotene content | *Dong et al. (2020)* |
| | *Wx1* | High amylopectin content | *Andersson et al. (2017)* |
| Sweet potato (*Ipomoea batatas*) | *IbGBSSI* | Low amylose content | *Wang et al. (2019)* |
| | *IbGBSSI, IbSBEII* | High amylose content | *Wang et al. (2019)* |
| Grapefruit (*Citrus paradisi*) | *ldnDH* | Low tartaric acid | *Ren et al. (2016)* |
| Mushroom | *PPO* | Anti-browning phenotype | *Waltz (2016)* |
| Soybean (*Glycine max* [L.] Merr.) | *GmFAD2–1A and GmFAD2–1B* | Improvement of seed oil composition | *Do et al. (2019)* |
| | *GmLox genes (GmLox1, GmLox2, and GmLox3)* | Breeding for lipoxygenase-free soybean varieties | *Wang et al. (2020)* |
| Cassava (*Manihot esculenta* Crantz) | *CYP79D1 and CYP79D2* | Production of Cyanide-free cassava | *Granada et al. (2020)* |
| | *GBSS and PTST/PTST1* | Production of waxy starch | *Bull et al. (2018)* |
| | *PTST1, GBSS* | Low amylose content | *Liu et al. (2021)* |
| Flax (*Camelina sativa*) | *FAD2* | Reduced polyunsaturated fatty acids | *Jiang et al. (2017)* |

**Table 2** (*continued*)

| Crop species | Target gene | Associated Trait | Reference |
|---|---|---|---|
| | GS9 | Slender grain shape | *Zhao et al. (2018)* |
| | OsGAD3 | High GABA content | *Akama et al. (2020)* |
| | OsNramp5 | Low Cd accumulations | *Tang et al. (2017)* |
| | OsFAD2-1 | High oleic acid proportion | *Abe et al. (2018)* |
| | OsPLD | Low phytic acid content | *Khan et al. (2019)* |
| Rice (*Oryza sativa*) | SlGAD2, SlGAD3 | High GABA content | *Nonaka et al. (2017)* |
| | OsGBSSI | Low amylose content | *Huang et al. (2020)* |
| | OsAAP6, OsAAP10 | Reduce GPC | *Wang et al. (2020)* |
| | OsBADH2 | Fragrant rice | *Ashokkumar et al. (2020)*; *Wu et al. (2018)* |
| | SH2, GBSS | Supersweet and waxy corn | *Dong et al. (2019)* |
| Carrot (*Daucus carota*) | DcMYB7 | Root color | *Xu et al. (2019)* |
| Kale (*Brassica oleracea*) | CRTISO | Yellow leaves and stems | *Sun et al. (2020)* |
| Banana (*Musa paradisiaca*) | MaACO1 | Long shelf life | *Hu et al. (2021)* |
| | OsGBSSI | Low amylose content | *Xu et al. (2021)* |
| | HvGBSSIa | Low amylose content | *Zhong et al. (2019)* |
| | Aux/IAA gene family (StIAA2) | Involved in Auxin/indole-3-acetid acid proteins synthesis | *Wang et al. (2015)* |
| | Acetolactate synthase1 gene (ALS1) | Biosynthesis of branched-chain amino acids | *Butler et al. (2015)* |
| Barley (*Hordeum vulgare L.*) | Granule-bound starch synthase (GBBS) | Involved in starch synthesis pathway | *Andersson et al. (2017)* |
| | Wx1 | High amylopectin content | *Satyawan & Santoso (2019)* |
| Rapeseed (*Brassica napus*) | BnITPK | Low phytic acid content | *Sashidhar et al. (2020)* |
| | BnTT8 | High oil production and GPC | *Zhai et al. (2020)* |
| Camelina (*Camelina sativa*) | CsFAD2 | High oleic acid proportion | *Khan et al. (2019)* |
| Banana (*Musa paradisiaca*) | MaACO1 | Long shelf life | *Hu et al. (2021)* |

(*Zhu, Li & Gao, 2020*), thus making it possible to domesticate new species within a short period (*Wolter, Schindele & Puchta, 2019*). Some recent studies have shown increased plant yield as a result of the manipulation of cytokinin homeostasis (among many factors impacting yield) as a practical technique to boost grain production (*Zhu, Li & Gao, 2020*). An instance of this is the wheat phenotype with high yields, created by knocking down the gene for cytokinin oxidase/dehydrogenase (CKX), an enzyme that catalyzes cytokinin degradation (*Nadolska-Orczyk et al., 2017*). While retaining the grain quality of rice, cultivars with higher tiller numbers and yields were created by knocking out the gene that encodes amino acid permease 3, which is important in nutrient partitioning. Furthermore, the terminus C (LOGL5), which encodes the activation of cytokinin enzyme in rice, has been edited for increased grain production in a range of climatic settings (*Zhu, Li & Gao, 2020*). Likewise, more complex traits have been edited in corn and tomatoes for greater productivity using CRISPR technology. The site-directed nucleases1 (InDel) was applied to generate some site-directed mutations in regulatory genetic regions controlling yielding in tomatoes, which affected their genetic variation and boosted their yielding in less time

**Table 3  The CRISPR-modified crops with improved growth and development.**

| Crop species | Target gene | Associated Trait | Reference |
|---|---|---|---|
| Wheat (Tritocum aestivum) | LAZY1 | Spreading of plant tillers | Miao et al. (2013) |
| | TaGW2, α-gliadin genes | Increase grain size | Sánchez-León et al. (2018); Wang et al. (2018) |
| | GW2 | Enhances grain weight | Zhang et al. (2018a) |
| Tomato (Solanum lycopersicum L.) | SP5G | Earlier harvest time | Soyk et al. (2017) |
| | fas, lc | Fruit size | Rodríguez-Leal et al. (2017) |
| | ENO | Fruit size | Yuste-Lisbona et al. (2020) |
| | CLV3 | Fruit size | Zsögön et al. (2018) |
| | GS3, Gn1a | Grain length | Shen et al. (2018) |
| | GW2, GW5, TGW6 | Grain length and width | Xu et al. (2016) |
| | GL2/OsGRF4, OsGRF3 | Grain size | Hao et al. (2019) |
| Rice (Oryza sativa) | GS9 | Slender grain shape | Zhao et al. (2018) |
| | GW5 | Grain width | Liu et al. (2017) |
| | OsGS3, OsGW2 and OsGn1a | Grain length and width | Zhou et al. (2019) |
| | Gn1a, GS3, DEP1 | Larger grain size, enhanced grain number, and dense erect panicles | Li et al. (2016) |
| Groundcherry (Physalis sp.) | ClV1 | Fruit size | Lemmon et al. (2018) |

compared to conventional breeding techniques (*Kawall, 2021*; *Tiwari, Singh & Behera, 2023*).

CRISPR-Cas technology has also been explored against plant biotic stress through the introduction of dominant resistant genes that could foster the development of resistance in pathogens (*Ahmad et al., 2021*). A popular example is the use of InDel in wheat to create a resistant variety against powdery mildew fungus, an obligate host-specific fungus called *Blumeria graminis* f. sp. *Triciti*. This fungus is known to be responsible for great losses of this crop across the world (*Lyzenga, Pozniak & Kagale, 2021*). It has also been applied to a promoter sequence of Argose (a maize line also called Zars) to confer a constitutive expression of the endogenous gene for improved massive yielding even under a drought-stressed environment (*Manna, Rengasamy & Sinha, 2023*). Another instance is the rice lines with broad-spectrum resistance to *Xanthomonas. oryzae* pv. oryzae created by employing CRISPR-Cas to modify the promoter region of *O. sativa* SWEET11, *O. sativa* SWEET13, and *O. sativa* SWEET14 (*Oliva et al., 2019*). Furthermore, a wheat cultivar with broad-spectrum powdery mildew resistance was developed by mutating all three mildew-resistance locus O (MLO) homologues EDR (encoding for enhanced resistance) to *B. graminis* f.sp. *tritici* at the same time (*Zhang et al., 2018a*). Moreover, mutagenizing the TaMLO-A1 and TaMLO-B1 genes with CRISPR-Cas9 resulted in bread wheat that is also resistant to powdery mildew (*Tyagi et al., 2021*). Similarly, tolerance to *Oidiumneo lycopersici*, which causes powdery mildew in tomatoes, was provided by targeting *Solanum lycopersicum* MLO1 with CRISPR–Cas in tomatoes. Defense against RNA viruses has been developed using the RNA-targeting Cas13a, Cas13b, Cas13d, and Cas12a *Francisella tularensis* subsp. *novicida* (strain U112) (*Wang et al., 2014*). Also, creating broad-spectrum viral resistance by knocking off plant susceptibility genes is a viable option that has already
been explored in Potyviruses of *Cucumis sativus*. By knocking down the eIF4E (eukaryotic translation initiation factor) gene in *Cucumis sativus*, which is not required for plant growth, a broad-spectrum resistance to potyviruses in cucumber was achieved without compromising its fitness (*Bastet, Robaglia & Gallois, 2017*).

## IMPACT OF CRISPR-CAS IN MANAGING THE ENVIRONMENTAL STRESS ON CROPS

Extreme temperatures (heat and cold), drought, salinity, UV radiation, and environmental pollution are some of the most prevalent abiotic stressors that have significant impacts on plants' yield, quality, growth, and development (*Wang, Vinocur & Altman, 2003*; *Li et al., 2023a*; *Li et al., 2023b*). Several structural and regulatory genes, together with non-coding RNAs, involved in crop response to varying environmental pressures have been targeted to improve crop tolerance to abiotic challenges (*Zhang et al., 2021*). The potential of the CRISPR-Cas system to increase the tolerance of plants to abiotic stressors was evident in the editing of the AGROS8 gene which is a negative regulator of ethylene response for enhanced maize tolerance to drought (*Hameed & Awais, 2021*). Similarly, SlAGAMOUS-Like 6 (SlAGL6) is an R gene that, when knocked out using CRISPR-Cas9, allows tomato plants to grow better and produce fruits even during heat stress (*Wan et al., 2021*). Furthermore, CRISPR-Cas-edited *Arabidopsis* mutants of dpa4-sod7-aitr256 improved plant tolerance to drought treatments, according to a recent study (*Zhang et al., 2021*). However, there are situations in which CRISPR-Cas9 was reported to show the opposite phenotype. This is the case for tomato plants where CRISPR-Cas9 knockdown of non-expressor of pathogenesis-related gene 1 (npr1) impaired drought tolerance (*Zhang et al., 2021*).

The challenges of climate change pose a major threat to food security and food safety as a result of the unpredictable weather patterns, rise in $CO_2$ concentrations, drought stress, disruption of plant defense mechanisms, and a spike in pests and pathogen populations and virulence (*Gangurde et al., 2019*). Moreover, for every 1 °C rise in global temperature, an estimated 10 to 25% of food crops, which include corn, rice, and wheat, will be damaged by pests. Thus, a drastic reduction is occurring in the expected yield, with an annual decline reported in some essential food crops such as; rice (0.3%), wheat (0.9%), and oil palm (13.4%) (*Basnet, 2012*; *Ray et al., 2019*). At the same time, total crop production needs to be doubled by 2050 to enable the global food system to support the growing population (*Pastor et al., 2019*).

Alleviating the effect of climate change on the quantity and quality of food products by using traditional approaches has not been really successful. In contrast, CRISPR-Cas systems have emerged as a precise, time and cost-effective method of adapting plants to meet future challenges. The technology was employed to improve the salt tolerance of rice by knocking out the OSRR22 gene, thereby making a crop that feeds 3.5 billion people across the world safe for consumption (*Zhang et al., 2019*). Also, CRISPR-Cas was used to boost maize grain yields under drought conditions by modifying the ARGOS8 gene, promoting cell division, and offsetting water scarcity (*Shi et al., 2017*). Furthermore,

CRISPR-Cas has been used to engineer disease resistance in tomatoes by inactivating a single gene (DMR6), which confers broad-spectrum disease resistance to the crop (*De Toledo Thomazella et al., 2016*). The mutants did not have any significant detrimental effects on growth and development while showing disease resistance against *Pseudomonas syringae, Phytopthora capsica*, and *Xanthomonas* spp. (*Kieu et al., 2021*).

In addition, herbicide-resistant weeds constitute a significant crop loss and pose a threat to global food security (*Manning & Soon, 2019*; *Tyczewska et al., 2018*). The production of herbicide-resistant crops using CRISPR-Cas has become an efficient technique for controlling weeds, and new technologies for transferring herbicide resistance to plants (*Han & Kim, 2019*). This is achieved by targeting endogenous genes like acetolactate synthase (ALS), cellulose synthase A catalytic subunit 3 (CESA3), 5-enolpyruvylshikimate-3-phosphate synthase (EPSPS), and splicing factor 3B subunit 1 (SF3B1) (*Hussain et al., 2021*; *Khan et al., 2022*). Herbicide tolerance can be achieved by the changes in the ALS gene of amino acid according to studies of naturally occurring point mutations in the ALS gene (*McNaughton et al., 2005*). In the report of *Preston et al. (2006)*, cytosine base editors (CBEs) were used to introduce specific base transitions into *O. sativa* ALS to confer herbicide resistance to rice while maintaining ALS activity.

Unlike the conventional breeding method, CRISPR-edited gene is time efficient and stable, as there is no distinguishable difference between the variants generated from genome editing and those obtained from naturally occurring variations (*Marone, Mastrangelo & Borrelli, 2023*; *Anders, Hoengenaert & Boerjan, 2023*). It was also more readily accepted in the market for commerce (*Brandt & Barrangou, 2019*).

## ADOPTION OF CRISPR-CAS: CURRENT POSITION AND IMPLICATIONS ON ETHICAL CONSIDERATIONS

About 190 million hectares of transgenic crops were grown during the past two years in 26 countries consisting of about 21 developing and five industrialized nations. Brazil, Argentina, and India are among the top five nations with the highest area of biotech crop production areas, accounting for 54% of the total growth in developing countries (*Turnbull, Lillemo & Hvoslef-Eide, 2021*). The industrialized nations which include the United States, Canada, Australia, Spain, and Portugal produce 46% of all biotech crops. However, governments and lawmakers across the world strive to safeguard their citizens, constituencies, and their environment, and this they do by ensuring the regulation of feeds, plants, and crops produced for consumption (*Hilbeck et al., 2020*). Variations occur in this regulation whether it is process- or product-oriented. Thus, the regulation of genetic engineering in general and the creation of transgenic organisms in particular is governed by two basic legal approaches (*Eckerstorfer et al., 2019*).

The product-oriented approach, which has historically been backed by the US, Canada, and other American countries, views genetically modified products or organisms (GM, GMOs) as the equivalents of those made by conventional selection processes (*Medvedieva & Blume, 2018*). The use of the aforementioned products or organisms therefore comes under the purview of current general legislation on the protection and eradication of

potential risks to individuals or the environment, and special regulations were therefore considered unnecessary (*Turnbull, Lillemo & Hvoslef-Eide, 2021*). On the other hand, the European Union has historically backed the process-oriented approach (*Medvedieva & Blume, 2018*). According to this perspective, gene modification technologies are unique and in particular quantitatively novel, hence the implementation of special regulations and legislation is thought to be essential (*Duensing et al., 2018*). The majority of the modified plant products, their production and release are already covered by European Union law particularly for some certain crops like corn, soybean, rapeseed and cotton. The law also permits various methods with adequate risk assessment measures with the exception of techniques such as mutation breeding used prior to the directive's implementation in 2001 (*Wolt et al., 2016*; *Van der Meer et al., 2023*). It could be possible to apply the present process-oriented European Union regulations to the CRISPR/Cas9 technology if it is considered a variation of the conventional genetic engineering that leads to the generation of GMOs (*Zhang et al., 2020*).

So far, more than 25 plant species and 100 genes have been successfully edited using CRISPR-Cas9 to achieve a range of desirable traits in important crops (*Manghwar et al., 2020*). Despite the fact that the CRISPR-Cas technology has been proven as a highly efficient genome editing tool for genetic improvement of crop production (*Zhang et al., 2020*), the controversy on the pros and cons of the consequences of its use in genome editing for agricultural food production was brought to the Court of Justice of the European Union (CJEU) in July 2018 (*Purnhagen & Wesseler, 2021*). The court, which issued a historic ruling, considered CRISPR-Cas as an application of 'mutagenesis exception' entrenched in Appendix 1 B of the Genetically Modified Organism (GMO) in the preliminary reference Confédération Paysanne (C-528/16) (*Siebert, Herzig & Birringer, 2022*). Hence, the new plant breeding techniques (NPBT) conclusively found 'oligonucleotide-directed mutagenesis' to be unqualified to receive mutagenesis exemption, after previously declaring the technology as a GMO-based technique (*Urnov, Ronald & Carroll, 2018*). However, such legal recognition of the regulation poses a disadvantage to the biotechnology enterprises that use the technology as a result of additional paperwork and expenses such as the requirement to acquire marketing authorization and to label the product among other things (*Turnbull, Lillemo & Hvoslef-Eide, 2021*). Thus, the acceptance and adoption of CRISPR-Cas as a new gene-editing technology has become a contentious and divisive social topic of international interest, especially among the scientists and the direct beneficiaries of the technology itself.

However, different states have different stances on this issue. An instance is the case of Argentina which in 2015 became the first nation to enact a unique regulatory legislation about novel plant breeding technology. This legislative document covered genome editing and declared that transgene-free goods were exempt from the GMO laws currently in effect (*Menz et al., 2020*). Given that the United States controls about 30% of the worldwide market for agricultural biotechnology, the nation is regarded as the world leader in the development and marketing of genetically modified crops. Contrary to most other nations, the United States lacks comprehensive federal legislation that regulates genetically modified organisms (*Wong & Chan, 2016*). Newly developed GM products are subjected to specialized regulatory authorities under the Coordinated Framework for Regulation

of Biotechnology (*Wolt & Wolf, 2018*). This entails that GM products are evaluated by the same health environmental and safety standards that apply to conventional products, which allows the designated authorities to treat similar products equally (*Turnbull, Lillemo & Hvoslef-Eide, 2021*). Similarly in Canada, a science-based assessment of risks that focuses on the product's allergenicity, toxicity, and off-target effects has been maintained (*Arpaia et al., 2020*). The regulations become effective when there occurs the trait expression of at least 20–30% lower or higher in a particular plant trait than in the conventionally grown varieties. Such a plant with novel traits is classified as a plant with new characteristics (PNT) and not a GMO (*Ferreira & Reis, 2023*). However, before any unconfined environmental releases can take place, the applications must pass through the Canadian Food Inspection Agency (CFIA) (*Smyth, 2019*). In Africa however, many countries including South Africa, Ethiopia, Malawi, Kenya, Nigeria Sudan, and Eswatini (Swaziland) are actively growing GM crops, despite the fact that population increase and climate change pose significant threats to food security (*Kedisso et al., 2022*). South Africa which is the producer of GM crops in the region was the first country in Africa to adopt a regulatory framework allowing GM crop cultivation, import, and export. This was followed by other listed countries and Burkina Faso which, although it has not commenced commercial production of GM crops, has had a Biosafety Law in place since the year 2012 (*Turnbull, Lillemo & Hvoslef-Eide, 2021*).

The need to further elucidate the safety of this system requires the evaluation of the possible constraints experienced in the adoption and use of this technology. This reveals two major concerns that could emanate from its use as off-target editing and unlawful or unethical scientific experimentation. As for the off-target editing, two possible impacts have been hypothesized. First, off-targets are expected in genomic regions that show high sequence similarity with the target. Second, off-targets are unexpected in genomic regions that are unrelated to the target (*Kaul et al., 2020*). As a result of the binding and cleavage that occur at sites other than the target DNA sequence, off-target editing can generate a loss of function in well-functioning genes or incorrect repair of disease-inducing genes thereby posing major therapeutic problems (*Klein et al., 2018*). Also, off-target editing can result in chromosomal rearrangements and other possible alterations that include the incorporation of DNA mismatches in the PAM-distal location of the sgRNA complementary sequence (*Manghwar et al., 2020*). Furthermore, off-target mutations caused by CRISPR-Cas9 may also have an impact on the edited organism and possibly its offspring (*Zhang et al., 2018b*), while future studies and advancements in this technology may reveal new unexpected off-target effects that could constitute unpredictable implications (*Moon et al., 2019*).

Despite the improved knowledge of CRISPR-Cas9 technology, a gap still exists in its targeting efficiency. The potential off-target effects have recently been a source of concern for CRISPR-Cas9 applications in plants, which needs to be addressed if the technology is to be widely used in gene therapy and crop breeding. However, the off-target effects are efficiently managed during the breeding process by the validation and the selection of phenotypes of interest to exclude off-target or inferior mutations that could result in inferior traits (*Bishop & Van Eenennaam, 2020*). Further, the current state of CRISPR-Cas9 applications in plant genome editing indicates that off-target mutations are uncommon

(*Wolt et al., 2016*). Unlike gene therapy and clinical research in humans (*Hunt et al., 2023*), plant research is not subject to the same ethical concerns, and off-target effects may be tolerated more readily.

## CONCLUSION AND FUTURE PERSPECTIVES

CRISPR-Cas is an essential technology due to its capability of rapid modification of plant genome to achieve the traits of interest while ensuring effective management of biotic and abiotic stressors. However, despite the enormous importance of CRISPR-Cas technology, it remains an object of debate over ethical issues and the safety of its global adoption and use. Scientific principles/application of the CRISPR-Cas system is not the main problem in crop production, but public acceptance and regulations for the products. New traits derived by conventional plant breeding are 'nature-identical', whereas the CRISPR technologies can be used for different purposes such as the development of exogenous genes and the propensity of such genetically modified organisms.

However, the basic expectation for the use of this technology should be more focused on the enhancement of 'nature-identical' traits. In addition, while the deployment of this technology in biomedical and human research could require strong ethical clearance, evidence shows that its application for agricultural purposes is relatively safe, especially in cropping agriculture. Nevertheless, we can say that despite the well-established use of the CRISPR-Cas tool for enhancing crop production at large, it is still not well explored due to its global regulations. In addition, more advanced CRISPR tools have been developed for the enhancement of the agricultural system over the last few years. These tools are beyond DSB-based editing, as they can recognize and target some specific portion of DNA sequences. DNA-targeting proteins such as the nuclease-dead Cas9 and Cas12a can be fused with domains carrying out different enzymatic activities. For example, dead Cas9 can be fused with the deaminase enzyme to enable the direct conversion of a single DNA nucleotide into a DSB. However, there is a limited availability of base editing platforms, namely C-T or A-G conversions, which has narrowed the sequence-editing windows. It is possible that broader platforms would emerge soon and remove this limitation. Then there will be a broader suite for the application of CRISPR tools such as the visual editing of specific genomic loci, the direct regulation of gene transcription, and the induction of targeted epigenetic modifications.

Taken together, there is a need for political willingness and public acceptance of the exploitation of this tool. Also, further research and observation of proactive events, which will facilitate CRISPR global acceptance needs to be carried out. Scientists should not discount the challenges and be transparent enough in providing CRISPR-breeding technology. This would go a long way in gaining the public and regulatory trust. The sustainable future of the crop system relies on the wide application of breeding tools that would be improved if used together with CRISPR tools. Moreover, resolving both the scientific and public/regulatory concerns would pave the way for sustainable crop production.

### Funding

Olubukola Oluranti Babalola was supported by the National Research Foundation, South Africa, for Grants UID123634 and UID132595 for this research. The funders had no role in study design, data collection and analysis, decision to publish, or preparation of the manuscript.

### Grant Disclosures

The following grant information was disclosed by the authors:
The National Research Foundation, South Africa: UID123634, UID132595.

### Competing Interests

The authors declare there are no competing interests.

### Author Contributions

- Akinlolu Olalekan Akanmu conceived and designed the experiments, performed the experiments, analyzed the data, prepared figures and/or tables, authored or reviewed drafts of the article, and approved the final draft.
- Michael Dare Asemoloye performed the experiments, analyzed the data, prepared figures and/or tables, authored or reviewed drafts of the article, and approved the final draft.
- Mario Andrea Marchisio performed the experiments, analyzed the data, authored or reviewed drafts of the article, and approved the final draft.
- Olubukola Oluranti Babalola conceived and designed the experiments, performed the experiments, analyzed the data, authored or reviewed drafts of the article, and approved the final draft.

### Data Availability

 This is a literature review.

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
