# Peer review of "Adoption of CRISPR-Cas for crop production: present status and future prospects"

_PeerJ, doi:10.7717/peerj.17402_

## Round 0.1 · original submission · Major Revisions

Dear Dr. Babalola,

Thank you for your submission to PeerJ.

Based on the perusal of review reports, and my own assessment; I feel that your manuscript needs to be thoroughly revised to make it suitable for publication. You have to carefully consider the reviewers' comments, and incorporate all the suggestions. You need to ensure that the revised manuscript reflects all the current global trends in the subject area so that it attracts a wide readership. In particular, a more extensive review of the literature needs to be carried out to capture the broader trends in CRISPR-Cas and present/critically discuss the same.

It is highly desirable that you get all the grammatical mistakes rectified, and improve the English language of the manuscript. I feel that Table 1 is too large; if possible you may split the information presented in Table 1 into two or three parts: in areas like plant growth and development; yield and quality, and biotic and abiotic stresses.

As abiotic stresses are an important theme of your paper, you are expected to provide some additional information as to how these stresses impact the global agricultural output, what are the shortcomings of conventional management options, and finally, how CRISPR-Cas could help in this regard.

We hope to receive the thoroughly revised manuscript within the stipulated time.

·

Basic reporting

There are many review papers on this aspects published in 2023.
Authors should include available updated information that is missing from this review.

These references and many more not covered by the Author topic is not thoroughly studied.
CRISPR/Cas9 therapeutics: progress and prospects

Nature Journal
https://www.nature.com › ... › review articles

by T Li • 2023



New advances in CRISPR/Cas-mediated precise gene ...

National Institutes of Health (.gov)
https://www.ncbi.nlm.nih.gov › articles › PMC10003097

by C Richardson • 2023 •



CRISPR-Cas: A robust technology for enhancing consumer ...

Frontiers
https://www.frontiersin.org › fpls.2023.1122940 › full

by V Verma — CRISPR-Cas is a versatile tool that can be used to knock out, replace and knock-in the desired gene fragments at targeted locations in the genome, resulting in ..



CRISPR/Cas genome editing in tomato improvement
https://www.frontiersin.org › fpls.2023.1121209 › full

by JK Tiwari • 2023 • Cited by 3 — In this mini-review, we provide a brief overview of CRISPR/Cas9, its current application in tomato trait modifications, and research advances on



Unintended CRISPR-Cas9 editing outcomes

Springer
https://link.springer.com › Human Genetics

by JMT Hunt • 2023 • Cited by 2 — Unintended CRISPR-Cas9 editing outcomes: a review of the detection and prevalence of structural variants generated by gene-editing in human ...



Using CRISPR/Cas to enhance gene expression for crop trait ...

Oxford University Press
https://academic.oup.com › jxb › article-abstract

by SS Ferreira • 2023 • Cited by 1 — Gene editing tools such as CRISPR/Cas9 are often thought of as a means to prevent gene expression; however, a more subtle and yet powerful



The CRISPR/Cas System: A Customizable Toolbox for ...

MDPI
https://www.mdpi.com › ...

by Y He • 2023 — This review discusses the potential and limitations of exploiting the sensitivity and specificity of detection techniques from laboratory to environmental ...

Exploring the potential of CRISPR/Cas genome editing for ...

John Wiley
https://onlinelibrary.wiley.com › doi › bit

In this review, we have introduced CRISPR/Cas9 and its applications in vegetable crops. We have also discussed the challenges and prospects of genome editing ...


Recent advances in CRISPR-based genome editing ...

BioMed Central
https://mmrjournal.biomedcentral.com › articles

by ZH Li • 2023 — After the CRISPR-Cas9 system was characterized and programmed to perform RNA-guided DNA cleavage at specific sites in prokaryotes, it was ..

Experimental design

Incomplete

Validity of the findings

Incomplete

Additional comments

Language needs plagiarism check and improvement.
Figures source should be mentioned i dont think it is prepared by author.
In current form It can not be evaluated.

Reviewer 2 ·

Basic reporting

Dear Author,
I have carefully reviewed the review article entitled “The future of food system: Adoption of CRISPR-Cas in Safe and Sustainable Crop Production” where I find the study a good research study towards food industry. Though studies has been reported till date regarding CRISPR Cas technology in crop production but close study with recent literature study will make one step forward towards advancement of functional and safe food community. But in this manuscript, I found the literature part of the MS inadequate in language framework therefore my only suggestion will be that to go thoroughly through the whole manuscript and write in a proper way. Kindly revise the manuscript carefully as there are some mistakes mentioned below which you need to rectify it.
1. Kindly arrange the keywords in alphabetical order
2. In line 54, sentence is not well framed
3. In line 59, Sentence is not well framed. It should be like achieving a secure and safe food system in the "time or hour" of an accelerating food demand will be appropriate.
4. As you have represented a workflow of CRISPR Cas in plants through graphical representation, it will be good if you write in words a little about its experimentation
5. In line 134-144, as you depicted figure 1 below Repetition of figure 1 here should be deleted
6. In line 146 As you have charted the table below in MS, repetition of table 1 here should be deleted
7. In line 167-168 Refer some references from where this statement was stated
8. In line 203 you have not cited any reference ,
9. In line 271-272 Is it? kindly cite some reference for proper validation
10. In line 288-289, By Other Americas you mean other American countries?
11. In line 298 Kindly write EU in full form
12. In line 298-300, Language is not well framed, kindly rewrite it for easy understanding
13. Some references are not in according to format, kindly go thoroughly and rectify it

Experimental design

no comments

Validity of the findings

no comments

Annotated reviews are not available for download in order to protect the identity of reviewers who chose to remain anonymous.

---

## Round 0.2 · Major Revisions

Dear Dr. Babalola

You are advised to go through the changes suggested by the reviewers in order to appropriately revise the manuscript. It is important to note that your revised manuscript will again be evaluated to ensure that you have properly addressed all the comments and suggestions.

Hope to receive the revised manuscript in due course.

Reviewer 2 ·

Basic reporting

Dear Authors,
I have carefully re-reviewed the MS entitled “The future of the food system: Adoption of CRISPR-Cas in safe and sustainable crop production”, where all the mistakes reviewed before has been addressed properly. But there are some minor mistakes which need to rectify such as
1)references cited in the literature part are not according to format ,
2)proper spacing in between some words are missing,
3)Scientific name of plant species are not in italics form
4)moreover references in the reference part are not according to the format e.g. some volume issue no. are in bold and some are not.

Experimental design

no comment

Validity of the findings

no comment

Reviewer 3 ·

Basic reporting

In Figure 1: the authors have depicted regenerated plants of some grass species/ family, and after that, they kept some broad leaf plant at final (f) that looks odd; therefore, keep the same plant species images for illustration of genome-edited plants.
The author also corrects F. Generation of to transgenic plants - Generation of transgenic plants. In the sub-heading, the authors have mentioned To, which should be modified into an appropriate title; the authors have skipped the generation of the null segregant part in genome-edited plants, which is an essential step in the genome-edited plants.
The author should add more published information about the genome-edited plant in the crop plants, including the field and horticultural crops.
L. 118. Italicize the Agrobacterium L233 Arabidopsis in the article
L. 241 (Gangurde et al., 2019). Moreover, for every 1 degree Celsius, write in better form one degree Celsius or 1° C.
The author should also elaborate on the global acceptance of genome-edited crop products and their exemption made by the different national and international regulatory authorities.

Besides, the prepared review article has met the professional English language requirement, sufficiently in the literature as references and within the scope of the journal.

Experimental design

Yes, the content of the prepared review article is within the aim and scope of the journal. In this review, the authors covered a substantial number of literature published in the public domain. However, the author missed the point of null segregant generation in the genome-edited and some elaborated points in the wide acceptance of genome-edited crop plants and products and regulatory exemption by authorities (from major biotech crop growing nations).

Validity of the findings

Yes, it is there in the review article.

Additional comments

NA

---

## Round 0.3 · Minor Revisions

Dear Dr. Babalola,

Thank you for your submission to PeerJ.

It is my opinion as the Academic Editor for your article - The future of the food system: Adoption of CRISPR-Cas in safe and sustainable crop production - that it requires a Minor Revisions, that is, in L248 oil palm dove, needs to be clarified during revision.

You are therefore advised to promptly address this suggestion and submit the revised draft ASAP.

Reviewer 3 ·

Basic reporting

The authors have substantially corrected the manuscript, and it may be accepted for publication. There is a one-line in L248 oil palm dove, which may be clarified during revision.

Experimental design
* * *
Validity of the findings
* * *
Additional comments

--

---

## Round 0.4 · Minor Revisions

Dear Dr. Babalola,

Thank you for your submission to PeerJ.

I am writing to inform you that your manuscript has in principle been accepted for publication but with the rider that the title needs to be changed a bit.

This is because the present title seems to be too bold. It could be changed, for instance, to "Adoption of CRISPR-Cas for crop production: present status and future prospects".

We apologise for this inconvenience.

I will expect to receive the manuscript with the revised title in a couple of days.

---

## Round 0.5 · Minor Revisions

Dear Dr. Babalola,

Please moderate the claim that "this method poses no safety risks"; in science, one must be very careful to claim absolute safety.

---

## Round 0.6 · accepted · Accept

Dear Dr. Babalola,

Thank you for your submission to PeerJ.

I am writing to inform you that your manuscript - Adoption of CRISPR-Cas for crop production: present status and future prospects - has been Accepted for publication. Congratulations!